# Ferroelectric Devices for Content-Addressable Memory

**DOI:** 10.3390/nano12244488

**Published:** 2022-12-19

**Authors:** Mikhail Tarkov, Fedor Tikhonenko, Vladimir Popov, Valentin Antonov, Andrey Miakonkikh, Konstantin Rudenko

**Affiliations:** 1Rzhanov Institute of Semiconductor Physics SB RAS, 630090 Novosibirsk, Russia; 2Valiev Institute of Physics and Technology RAS, 117218 Moscow, Russia

**Keywords:** ferroelectric, memristor, FeFET, FTJ, content-addressable memory

## Abstract

In-memory computing is an attractive solution for reducing power consumption and memory access latency cost by performing certain computations directly in memory without reading operands and sending them to arithmetic logic units. Content-addressable memory (CAM) is an ideal way to smooth out the distinction between storage and processing, since each memory cell is a processing unit. CAM compares the search input with a table of stored data and returns the matched data address. The issues of constructing binary and ternary content-addressable memory (CAM and TCAM) based on ferroelectric devices are considered. A review of ferroelectric materials and devices is carried out, including on ferroelectric transistors (FeFET), ferroelectric tunnel diodes (FTJ), and ferroelectric memristors.

## 1. Introduction

Conventional von Neumann architectures suffer from long latency and high power consumption due to data movement between external memory and arithmetic logic units (ALUs) [1,2,3]. In-memory computing [4,5,6,7,8,9,10,11,12,13] is an attractive solution for reducing power consumption and memory access latency cost by performing certain computations directly in memory. Content-addressable memory (CAM) is an ideal way to smooth out the distinction between storage and processing, since each memory cell is a processing unit. CAM [14,15,16,17,18] compares the search input with a table of stored data and returns the matched data address. CAM modules have a higher throughput than other hardware and software search engines. CAM can be used in a wide variety of applications requiring high search speed.

The main commercial application of CAM today is the classification and forwarding of Internet Protocol (IP) packets in network routers [19,20,21,22,23,24]. CAM is a good choice for implementing fast searches. However, the CAM speed comes at the cost of increased silicon area and power consumption, two design parameters that developers seek to reduce. As the number of applications that require a larger CAM grows, the power issue becomes even more acute. Reducing power consumption without sacrificing speed or area is a major focus of recent research in high capacity CAM.

Figure 1 shows a simplified CAM block diagram. The system input is a search word which is translated through the search strings into a table of stored data. Each stored word has a match string that indicates whether the search word and the stored word are identical (match case) or different (mismatch case or miss). The match lines are passed to the encoder, which generates a binary match location corresponding to the match line in the match state. The encoder is used in systems where only one match is expected.

The CAM general function is to take the search word and return the corresponding memory location. This operation can be thought of as a fully programmable arbitrary mapping of a large input search word space to a smaller output match location space. 

Ternary content-addressed memory (TCAM) [24,25,26] uses the “don’t care” state in addition to the “0” and “1” states for wildcard operations that match both “0” and “1”. This is implemented by making it possible to always disable the discharge path in the TCAM cell (Figure 2) (typically by turning off the corresponding switches controlled by the “don’t care” bits). As a result, the TCAM cell requires 3-state memory: ‘0/1’, ‘1/0’, and ‘0/0’, and is typically implemented with two bits (‘1/1’ is redundant and not used). With a TCAM-based approach, given a query feature vector, it is possible to search quickly and efficiently across all stored vectors (i.e., network storage devices).

TCAM performs lookup operations directly on the memory itself. TCAM is a critical component for achieving fast lookup. TCAM performs a bitwise XOR/XNOR between the lookup key and all the stored data to obtain the result of a match in one cycle. CAM or TCAM is usually organized as a two-dimensional array. The input vector bitstream is sent to the array via the vertical search lines SL. The corresponding input bit and the stored bit in each CAM cell perform an XNOR-style operation that resets ML if two bits do not match (or leaves ML floating, otherwise). Therefore, if one or more cells do not match, ML will be sparse; otherwise, ML will remain at a high level and give a “consistent” result.

## 2. Ferroelectric Materials and Devices 

Today, the main type of manufactured electronic devices are integrated circuits of computing systems based on complementary metal-oxide-semiconductor (CMOS) silicon transistors [27,28,29,30]. Processor static cache and DRAM circuits use the most advanced 5–10 nm design rules, while SoC and flash non-volatile memory use the less sophisticated 10–14 nm and 22–28 nm, respectively. 

The highest demands on memory capacity and speed are made today by neuromorphic computing systems. Despite significant progress in the systems software implementation based on mathematical models of synapses and neurons, the hardware approach is of considerable interest, since it will allow the implementing of the fastest and most energy-efficient approaches to solving a number of artificial intelligence (AI) problems. Therefore, the study of the physical foundations and technologies for the formation of integrated universal memory devices, with the possibility of carrying out data operations and functionally combining the best characteristics in terms of capacity, speed and energy efficiency of all memory devices types, remains an urgent problem in semiconductor micro- and nanoelectronics. Ferroelectrics belong to the multifunctional materials class integrated into CMOS technology.

### 2.1. Ferroelectric Materials 

Two stable polarization states of ferroelectric materials (FM) can be switched by an electric field application [30,31]. FM has a stable polarization at zero applied field, called remanent polarization *Pr*. All FM are piezoelectrics [32]. As a result, FMs have many useful properties [33,34]. When it comes to the use of ferroelectrics in integrated circuits, the defining ferroelectric property, namely the switchable polarization *P*, is the most important property, since it can be applied to information storage, for example, in a ferroelectric capacitor [34] (Figure 3a). Here, the field-driven switching mechanism, together with the fact that the polarization state will persist for a long time, makes the material an ideal choice for implementing low-power non-volatile memory.

It is believed that, for ferroelectricity realization in a crystal, it is necessary to have a non-centrosymmetric structure. This makes ferroelectricity a rare material property that has historically been observed only in fairly complex crystal structures involving three or more elements, such as perovskites. However, they are difficult to use in integrated circuit (IC) technology, which places high demands on the heat budget, exposure to forming gas during annealing, and tight control of the elements used in the production line. As a result, ferroelectrics integrated into semiconductor ICs so far hold a small market share.

The first ferroelectrics, discovered in the 1920s and 1930s, respectively, were crystals of water-soluble Rochelle salt and potassium dihydrogen phosphate [35,36,37]. The discovery of ferroelectricity in barium titanate BaTiO_3_ (BTO) in the early 1940s paved the way for applications in sensors and capacitors [38,39,40]. As far back as the 1950s, Buck [41] proposed the use of ferroelectrics in memory applications, and this proposal inspired additional research [42,43]. However, this work did not bring products to market, and IC technology was not yet available in those days. However, early devices that contained hundreds of capacitive crossbar array memory cells on a single BTO substrate can be considered as the first integrated memory circuits demonstration.

Today, similar matrices are formed on the basis of memristors that provide hardware implementation of neuromorphic computing systems. One of the key elements of this approach is the perovskite memristor, a bipolar device with a continuous spectrum of possible resistance values, capable of modeling synaptic plasticity [44,45]. In the early 2010s, a resistive switching was shown due to the conductive filament formation in resistive random access memory (ReRAM) cells with a metal–insulator–metal Hf/HfO_2_/TiN stack which was actively introduced into CMOS technology [46,47,48,49]. As for other materials (Figure 4, Table 1), the lead–zirconium titanate Pb[Zr_x_Ti_1−x_]O_3_ (PZT) system took another big step in the 1950s [50,51].

As with BTO, PZT has a perovskite structure. However, a mixture of oxides, based on Zr and Ti, provides additional flexibility, and has excellent ferroelectric properties. In the second half of the 1950s, the concept of a ferroelectric field-effect transistor (FeFET) was first proposed [52]. However, it took a very long time before such a device gave useful characteristics, including non-volatile storage of information [53]. FM integrated circuit realization could meet all the requirements and be commercialized in the early 1990s [54].

Aside from using PZT instead of BTO, the main difference from the earlier 1950s attempts was that a selecting transistor was added to the memory cell, resulting in a cell structure similar to dynamic random access memory ICs (DRAM). The sampling transistor eliminated the problems associated with accessing other columns of bits (bitline) and lines of words (wordline). However, PZT capasitors were subject to material fatigue degradation, which manifested itself in a decrease in switchable polarization with an increase in the number of read and write cycles. 

In the 90s, layered perovskites were proposed to solve fatigue problems, with oxide dielectric interlayers between the perovskite layers, such as strontium–bismuth tantalate Sr2Bi2TaO9 (SBT), as a solution to the fatigue problem (Figure 4) [55].

Commercial success was limited, since around the same time it was found that fatigue in PZT was significantly reduced when using oxide electrodes such as IrO_2_, RuO_2_ and related materials [56]. Even after solving the fatigue problem in PZT, the required crystallization and crystallite size, unfriendly inclusion of lead and weakly bound oxygen, low coercive field, and high leakage currents made perovskites and layered perovskites quite problematic for integration into CMOS processes [57]. As a result, the most advanced technology based on perovskites has stopped at the 130 nm design rule [58], and the problem of three-dimensional integration has not yet been solved [59]. It was not until the mid-2000s that the first demonstration of the energy-independent operation of SBT-based FeFETs was achieved [60]. In the same time period, a concept based on switched tunneling current through a very thin ferroelectric, which was first proposed by Leo Esaki et al. back in 1971 [61], was finally implemented [62] by adding ferroelectric tunnel junctions (FTJs) as a fourth option in the portfolio of ferroelectric storage devices. However, FTJ devices require high-quality epitaxial ferroelectrics [63].

It is unlikely that this approach will be integrated into the CMOS process. Epitaxial growth directly on a CMOS-compatible substrate cannot be achieved with currently known processes for the epitaxial growth according to the lattice parameters. An alternative approach for integrating such complex oxide films and metal electrodes is to grow epitaxial films on another substrate and then transfer them to CMOS, but this method is still at the stage of fundamental studies of wafers [64,65]. It will take many more years of research to understand if an FTJ device can be viable for CMOS integration. Thus, by the end of the first decade of this century, the field of ferroelectric devices was demonstrated for four fundamentally different types of devices, but their integration into modern CMOS processes was hampered by the problems of rather difficult to manufacture materials that are incompatible with the integrated circuit CMOS technology. 

In 2011, it was first reported that a ferroelectric effect can be achieved in doped hafnium oxide (HfO_2_) [66,67]. This discovery changed the prospects for integrating ferroelectrics into complementary metal–oxide–semiconductor (CMOS) processes, even those that had been at the front end of CMOS processes since 2007 [68]. Quite recently, piezoelectricity used in AlN-based devices has been successfully converted into switchable ferroelectricity in AlScN [69,70]. This material would be ideal for CMOS integration with high power Al_x_Ga_1-x_N nitride technology. Finally, efforts to create 2D materials suitable for use in electronic devices have also led to very interesting ferroelectrics [70,71]. In the next section, ferroelectrics based on hafnium oxide will be considered, since these materials are the closest to real applications and, therefore, show promise for implementation in commercial ferroelectric memristors, tunnel junctions, and transistors in the next five years.

### 2.2. Ferroelectric Materials Based on Hafnium Oxide 

A possible way to solve the problem of integrating ferroelectrics into CMOS IC technology was to detect the ferroelectric properties of 10–15 nm films based on hafnium dioxide (HfO_2_) doped with Si (HO:Si) in a coercive field Ec = (1 − 2) × 10^6^ V/cm group from the Technical University of Dresden (Germany) in 2011 [66,67]. Since undoped HfO_2_ (HO) films have been actively used since 2007 as gate high-k insulators in all modern CMOS ICs with supply voltages up to 1.5 V [68], there are no technical barriers to the introduction of ferroelectric HfO_2_ layers doped with CMOS IC technology. A little later, the same group demonstrated a decrease in Ec and an increase in the residual polarization P_r_ to 10–20 µC/cm^2^ in a solid solution of hafnium–zirconium dioxide (Hf_0.5_Zr_0.5_O_2_ or HZO) [72]. 

The authors of the first works with electron and X-ray diffraction methods showed that the ferroelectric hysteresis in HO:Si and HZO layers is due to the metastable non-centrosymmetric orthorhombic phase Pca2_1_ (Table 2). This phase in HO:Si and HZO films was transformed into a stable monoclinic phase P21/c during long-term stationary heat treatments (FA) above 600 °C and 500 °C, respectively [73]. At the same time, the temperature budget for manufacturing modern CMOS ICs reaches 900–950 °C.

It was proposed to solve the problem of thermal stability of ferroelectric layers with metastable ferroelectric phases in three ways: firstly, by using rare-earth impurities (La, Gd, Y, and others) or Al for doping with HO instead of silicon [74,75,76], which form refractory oxides due to the greater bonding energies; secondly, by a decrease in the duration of the thermal load on the ferroelectric during fast (sub-minute) heat treatments (RTA) [77]; thirdly, instead of introducing disordered inclusions of refractory oxides, it was proposed to split films with a thickness of more than a few nanometers into nanometer layers due to regular insertions of Al_2_O_3_ monolayers separating thick films into nanometer HO or HZO lamellae. Such films, nanolaminates, hereinafter referred to as HAO or HZAO, respectively, ensured the expansion of the thermal stability region of ferroelectric phases up to temperatures RTA T ~ 900 °C while maintaining the maximum value of the remanent polarization P_r_ together with the use of the first two methods simultaneously [78,79,80,81,82,83]. Such inserts slow down the accelerated growth of a stable monoclinic phase in nanolaminates during RTA heat treatments due to the size effects of the contribution of surfaces to the free energies of nanosized nuclei of all phases formed during plasma-assisted atomic layer deposition (PEALD), taking into account higher crystallization temperatures of amorphous Al_2_O_3_ inserts compared with HO and HZO [84].

### 2.3. Ferroelectric Transistors

Four promising types of non-volatile memory devices based on ferroelectric materials have already been proposed, where the ferroelectric tunnel junction and the memristor are combined (Figure 1c). So far, the only commercially successful concept is ferroelectric random access memory (FeRAM). FeRAM devices demonstrate a high read and write speed (~10 ns), a giant resource (~10^14^ switchings), and have already found their niche in the market of modern non-volatile memory [54,57,58,60]. 

In FeRAM 1T1C cell, during a destructive read, the polarization-switchable charge is transferred through the sampling transistor to the discharge line (BL) (Figure 3a). Just as with DRAM, the sensitivity amplifier determines the stored logic state. The read cycle must be completed by restoring the information. Therefore, each read cycle also increases the write time. Using PZT as a ferroelectric, it is possible to achieve a cycle life of ~10^6^ switchings. For better scalable films based on hafnium oxide, a resource reaching 10^12^ switching cycles has been observed, on which successful implementations of 1T1C memory arrays have recently been demonstrated [85,86]. In addition, the low charge signal can be sufficiently amplified by a second transistor, which is added to the memory cell, forming a 2T1C cell. This reduces the complexity of manufacturing a planar capacitor to a minimum, however, by increasing the size of the memory cell [87].

Further development of FeRAM will focus on scaling and 3D integration [88,89], as well as optimizing the reliability of ferroelectric capacitors, which is a mandatory requirement in the manufacture of memory arrays with a density of several Gbits. However, FeRAM arrays are not without a number of significant drawbacks. Since the state of the FeRAM memory cell is directly encoded by the direction of the remanent polarization vector, reading this state is destructive and requires restoring information that slows down reading and increases power consumption.

The actively developed 1T memory cell in the FeRAM architecture, which does not have such disadvantages, uses a ferroelectric field effect transistor (FeFET) with a ferroelectric instead of a gate dielectric. The FeFET idea is not new. It was proposed back in 1957 [52], but was commercially implemented only on the basis of SBT ferroelectric in 2004 [60]. FeFET research has made significant breakthroughs over the past decade. FeFET is a field effect transistor in which the direction of the remanent polarization vector modulates the conductance of the transistor channel. The key parameter of a ferroelectric field effect transistor is the memory window (Memory Window, MW). The value of the MW is determined by the region of gate voltages at which the two states of the channel resistance differ by more than an order of magnitude. This region is determined not only by the magnitude of the remanent polarization, but also depends on the magnitude of the coercive fields in the ferroelectric. For example, if the coercive field is small, as in the case of perovskites, then it is necessary to use large thicknesses of the ferroelectric layer. Otherwise, even small gate voltages will switch the polarization of the ferroelectric. This effect limited further scaling of FeFET devices based on classical ferroelectrics [52]. Today, this limitation has been circumvented by using ferroelectric hafnium oxide which, on the contrary, has large (0.8–2.0 MV/cm) coercive fields. As a result, a record channel length of 22 nm was achieved in HfO_2_-based ferroelectric field-effect transistors [89], but FeFET simulations with HZO ferroelectric parameters demonstrate its applicability in the most advanced 7 and 3 nm process standards [90].

In the 1T FeFET concept, using the internal amplification of the transistors, a non-destructive reading can be achieved by measuring the drain-source current, while the data stored in the gate ferroelectric can be maintained in a non-volatile manner. Successful integration of hafnium oxide FeFET into commercial 28 nm and 22 nm planar technology has been demonstrated with modern high quality metal gate technologies [89,91]. Today, the cyclic resource is still on the verge of what is required for non-volatile memory, mainly due to the degradation of the interfacial layer that forms between the Si channel and the ferroelectric layer [92]. However, the claimed 10^10^–10^12^ write cycle life is already competitive, with conventional floating gate and charge trapping devices, making the FeFET device a suitable solution for embedded non-volatile memory (eNVM). Further improvement in cycle life can be achieved by eliminating the interfacial oxide layer, or by designing a device structure that changes the capacitive voltage divider between the ferroelectric and dielectric layers [92,93]. In terms of commercialization, basically three reasons are decisive. First, from the analysis of array distortions, array architectures such as OR and NOR are preferred, implying a larger cell size compared to NAND [94]. Secondly, the polycrystallinity of the ferroelectric material causes a certain variability in switching characteristics [68], which is usually compensated by an increase in the size of the transistor beyond 0.01 μm^2^ [89,91] or a change in the FeFET design, which still prevents the FeFET concept from being scalable to competitive sizes, for energy independent storage devices. An improvement can be achieved by using textured or epitaxially grown ferroelectric films. Thirdly, the storage of several bits in a single FeFET was successfully demonstrated [95]. However, in the case of FeFET, multi-level storage is limited by the maximum achievable threshold voltage (MW) memory window calculated from a first order estimate over MW = 2 * E_c_ * t_ox_ ≈ 2 V, and implementation of more than 2 bits per cell seems unrealistic. The way to increase MW is to increase the thickness of the FE layer, as shown in reference [96]. Another proposed way to increase the number of bits is to change the FeFET design to improve the field uniformity in the ferroelectric along the channel (Figure 5), or to use two or more gates [97,98,99]. 

Scalability limitations can be overcome by implementing high-density 3D-NAND FeFET memory [88,100,101]. In addition to the larger effective gate area of the FeFET, which would improve layered memory over planar FeFET, improved die efficiency could be achieved by lower on-chip software voltages for FeFET compared to flash devices. However, the cyclic resource can also be a decisive factor here [102]. Devices based on non-volatile memory in fluorite-structured oxides, such as HfO_2_ and ZrO_2_, are able to mimic the behavior of synapses and neurons, which are the main elements of biological computing systems (the brain). With the discovery of ferroelectric devices compatible with modern CMOS technologies (Figure 6), neuromorphic computing devices based on ferroelectrics with a fluorite structure quickly became the main direction of neuromorphic research and development [73]. Additional opportunities for neuromorphic computing are opened by the analog operation mode of FeFET in the architecture of analog computing in memory (analog compute-in-memory, ACiM) [103].

### 2.4. Ferroelectric Tunnel Junction

A ferroelectric tunnel junction (FTJ) was proposed as a non-volatile memory relatively recently, and the possibility of its fabrication based on ferroelectric hafnium oxide was confirmed only a few years ago [55]. The classic ferroelectric tunnel junction is a two-electrode device in which the direction of polarization modulates the potential barrier through the tunnel-transparent ferroelectric, and, as a result, the tunnel current flowing (Figure 7). 

The current measured at different directions of the polarization vector is a logical “0” and “1”. The main advantages of FTJ are not limited to two-electrode control and non-destructive reading of the memory state, which can significantly increase the recording density in the crossbar architecture of memory arrays, even compared to classical DRAM memory [93].

Unshielded surface charges induced by remanent polarization create a depolarizing field. The depolarizing field changes the resulting potential barrier across the structure. A change in charge signs during polarization reversal also changes the direction of the depolarizing field and the profile of the potential barrier. As a consequence, a change in the potential profile upon polarization inversion is possible only with nonideal screening of surface charges, which, in turn, is ensured by a finite screening length in the electrodes and (or) the presence of an IL dielectric. Changing the potential barrier with a change in polarization will lead to a change in its transparency and a change in the tunneling current (Figure 6A–C) [73]. The change in current can be enhanced in the case of an asymmetric potential profile by using electrodes with different screening lengths, for example, in a metal-ferroelectric-metal or metal-ferroelectric-semiconductor structure [104,105,106]. The screening length in metals is small, so the polarization charges are screened almost completely, creating a small depolarizing field. The presence of a dielectric layer (IL) or a semiconductor as one of the electrodes weakens the shielding of polarization charges and introduces additional asymmetry into the structure, which makes it possible to more effectively modulate the potential distribution by rotating the polarization vector [107,108,109,110]. The presence of a space charge in the ferroelectric layer creates a number of effects common to all ferroelectric materials. Space charges on traps and mobile charged defects in a ferroelectric determine the imprint (switching voltage drift) and switching kinetics, and also lead to a gradual evolution of the domain structure and a decrease in the residual polarization during cycling [73,111,112,113].
Figure 7Energy band landscapes for different structures with different ferroelectric polarization states within the structure. (**A**) Metal/ferroelectric/metal (M/F/M), (**B**) metal/ferroelectric/interlayer/metal (M/F/IL/M), and (**C**) metal/ferroelectric/IL/semiconductor (M/F/IL/S) structures. The band structures are drawn using band diagram program with schematic Thomas–Fermi screening length of metals [113,114]. The interlayers of M/F/IL/M and M/F/IL/S are set as Al_2_O_3_ and SiO_2_, respectively. (**D**) Conductance ratio of FTJs, having a TiN/Hf_0.5_Zr_0.5_O_2_ (HZO)/ZrO_2_/TiN (M/F/IL/M) and TiN/HZO/ZrO_2_/poly-Si (M/F/IL/S) structure as a function of the pulse amplitude. (**E**) Retention characteristics of M/F/IL/M and M/F/IL/S FTJs; both devices were measured after wake-up field cycling of 10^6^ cycles with ±6 V/500 ns and 10 V/500 ns for M/F/IL/M and M/F/IL/S, respectively. (**F**) Relative permittivity of a M/F/M capacitor as a function of the field cycles. The inset shows the device schematic (**right**) and a hysteresis loop of the relative permittivity (**left**). The relative permittivity was extracted from the small-signal capacitance measured at 0 V with a bias amplitude of 30 mV. The program/erase pulse amplitude used were 3 V/−3 V. (**D**,**E**) Reproduced with permission [104]. Copyright 2021, IEEE. (**F**) Reproduced with permission [115]. Copyright © 2021, IEEE [73].
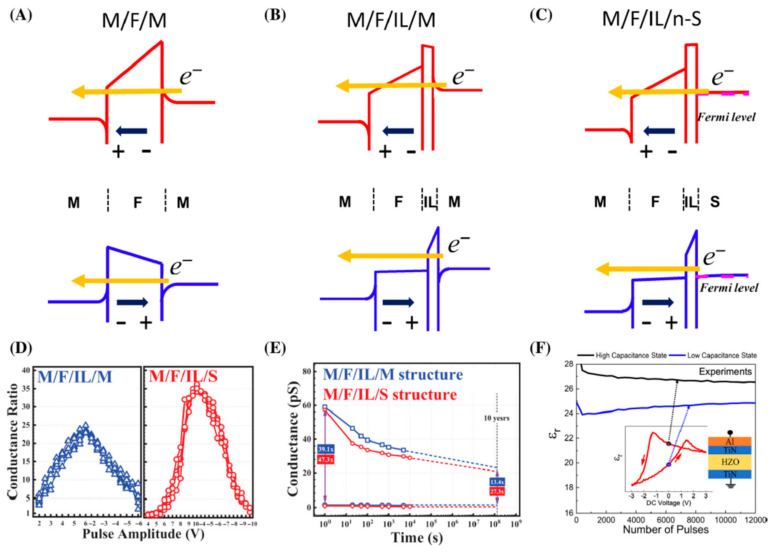



### 2.5. Ferroelectric Memristors

Currently, a large number of memristive devices based on various switching mechanisms have been demonstrated. In memristive devices, a change in resistance is associated with a change in the activity of a biological synapse (synaptic weight). Based on the principle of operation, they can be divided into two classes: memristors of the first and second kind [114,115,116,117,118,119,120]. In memristors of the first kind, conductivity modulation is carried out only by changing the external applied voltage, as a result of which the demonstration of synaptic properties is possible only when using overlapping impulses (post- and pre-synaptic spikes) of a complex shape. 

The biological synapse has an internal dissipative temporal mechanism that determines its current state and synaptic plasticity. The synaptic response to a series of coding spikes is determined by many temporal processes in the presynaptic and postsynaptic neuron, among which are, for example, the diffusion of Ca^2+^ ions through the postsynaptic membrane [121], the concentration of which gradually decreases with time and strongly depends on the prehistory. All these mechanisms lead to the fact that the synaptic response depends on the frequency of the stimulating impulse, rather than its shape. 

Memristors, in which similar dissipative mechanisms are observed, do not require accurate tuning of stimulating impulses and their overlap, and therefore can be used for “natural” modeling of the activity of a biological synapse (Figure 6). Such devices are called memristors of the second kind. In the practical implementation of memristors of the second kind, the dissipation of oxygen vacancies in the conducting channel [119], a decrease in their mobility [120], or the minimization of the surface energy by hydrogen and metal ions [122] act as an internal time mechanism. The stochastic nature of ion migration results in these devices having poor repeatability and a low number of possible switches.

Memory devices based on ferroelectric materials, which have reproducibility and a cyclic resource, are a more promising direction for creating memristors of the first and second kind. The ferroelectric memristor [121,123] is a natural development of the FTJ idea. However, instead of two possible states, these devices use one more degree of freedom associated with the domain structure of the ferroelectric film, resulting in an almost continuous spectrum of possible resistance values. Since switching in these devices is completely determined by the change in polarization direction, a very interesting feature of ferroelectric memristors is that the dependence of the R_OFF_/R_ON_ ratio on the switching voltage is similar to the shape of the hysteresis loop curve (Figure 8). It should be emphasized again that this correlation is an excellent criterion for verifying the ferroelectric nature of switching observed in a particular device. Until recently, research in this area was focused on ferroelectric memristors of the first kind, including memristors based on ferroelectric hafnium oxide [112,124]. A ferroelectric memristor of the second kind was first demonstrated in [125].

## 3. Content-Addressable Memory Based on Ferroelectric Devices

In recent years, ReRAM [126] has become a potential non-volatile memory (NVM) candidate for a next-generation storage system with high read/write speed, low programming voltage, and good scalability. The ReRAM device is usually a three-layer device formed by a metal-insulator-metal package. It can switch from a high resistance state (HRS) to a low resistance state (LRS) with a SET operation, and from LRS to HRS with a RESET operation. ReRAM-based CAMs [126,127,128,129] typically consume higher power during write operations due to their device state switching mechanisms. In addition, ReRAM’s low on-off ratio also results in high detection complexity and cost, since multiple NVM-based CAM cells in the off state can sum up significant discharge current close to on-state currents. These new devices suffer from significant on- and off-state current variations, limiting scalability and reliability. Ferroelectric-based TCAMs are much more compact [130].

A single TCAM cell based on CMOS requires 16 transistors (Figure 9a), occupies a large area, and is volatile, and thus incurs a footprint and leakage energy penalty. BL stands for bit line, SL for search line and WL for write line. TCAMs based on resistive storage elements (Figure 9b) reduce the cell footprint, but suffer from high write energy, low ON/OFF ratios and variation issues. TCAMs based on four CMOS FET and two FeFETs are more energy efficient. TCAMs based on only two FeFETs (Figure 9d) are non-volatile, and achieve the best energy and latency product performance [131]. TCAM cell parameters are shown in Table 3.

FeFETs are in fact MOSFETs with a ferroelectric layer integrated into the gate stack [130,131,132,133,134,135,136,137,138,139]. The interaction between the ferroelectric layer and the gate oxides of the MOSFET results in unique FeFET characteristics. FeFET stores the polarization direction in the ferroelectric layer as a memory state. The direction of polarization changes the threshold voltage V_TH_ FeFET from low to high. It should be noted that the low-level V_TH_ can be set to either a negative or positive value to provide low or high channel resistance with zero gate voltage V_G_. 

Writing in FeFET transistors can be accomplished by applying a voltage to the ferroelectric layer that exceeds the coercive voltage for a certain period of time. In general, a positive (or negative) voltage applied to the gate of an n-type FeFET tends to decrease (increase) the V_TH_ of the device. Polarization switching can be modulated by adjusting the amplitude or duration of the write voltage pulse applied to the gate. A FeFET read can be carried out by detecting the drain current with an applied gate voltage lower than the write voltage to avoid disturbing the read. FeFETs based on hafnium are highly scalable. Reports indicate that FeFETs can exhibit high on/off resistance ratios in excess of 10^6^, implying large memory arrays can be used. FeFETs also exhibit high switching speed, moderate life and moderate write voltage. FeFETs have a high on/off ratio, which is good for scalability. FeFETs do not consume DC power during write and seek operations, which also results in high energy efficiency. The FeFET TCAM design in Figure 9c additionally uses FeFET as both memory and comparator, resulting in very high TCAM density. The more bit mismatches, the faster the ML discharges.

Hafnium-based oxides can be used in FeFETs or tunnel junctions (FTJs) [135,136] using the HfO_2_ ferroelectric phase. The threshold voltage of FeFET, and hence its drain current, is controlled by the polarization state of the gate dielectric, which can be controlled by appropriate positive and negative values of the gate voltage pulses. Of particular importance for neural network training, FeFETs provide faster writes at a lower voltage than the conceptually similar flash memory. The service life is currently limited to approximately 10^6^–10^10^ cycles, and improvements require devices, materials, and technological innovations [136]. Based on the SPICE model of double-gate FeFET (2G FeFET) [138], a crossbar model was built that can be used as CAM [139]. 

Alternatively, ferroelectric tunnel junctions (FTJ) [140] metal-ferroelectric-metal (MFM) can be used, since MFM devices have greater cyclic stability than FeFETs. The HfO_2_ based FTJ has been developed as a promising non-volatile storage device with high CMOS compatibility, scalability, low power consumption and non-destructive readout. In FTJ, the change in polarization caused by an electric field applied across two electrodes modulates the potential barrier, causing the tunneling electrical resistance effect. FTJ-based switch point arrays with the size of a memory cell have the advantage of high integration density compared to arrays based on 3-terminal FeFET devices.

## 4. Conclusions

The issues of constructing binary and ternary content-addressable memory (CAM and TCAM) based on ferroelectric devices are considered. A review of ferroelectric materials and devices is undertaken on ferroelectric transistors (FeFET), ferroelectric tunnel diodes (FTJ), and ferroelectric memristors.

## Figures and Tables

**Figure 1 nanomaterials-12-04488-f001:**
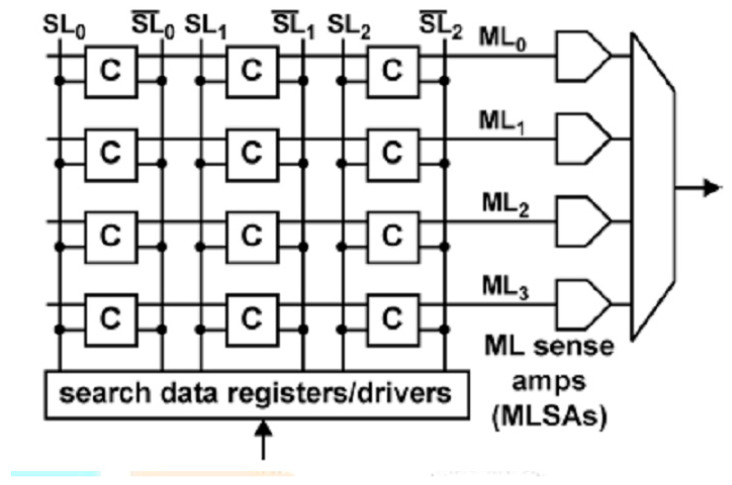
CAM block diagram [18].

**Figure 2 nanomaterials-12-04488-f002:**
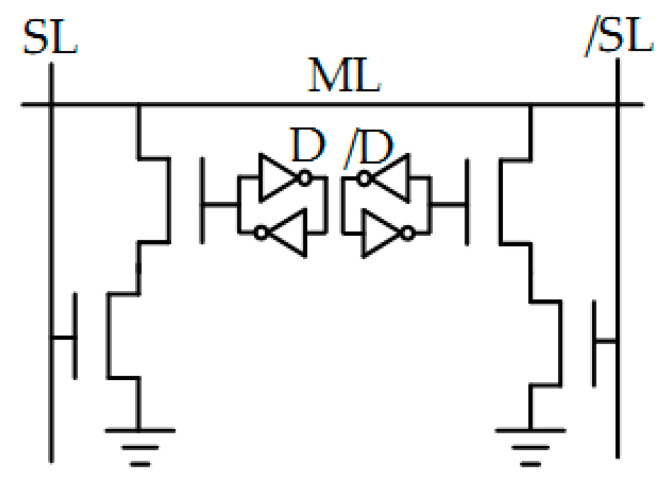
TCAM cell [24].

**Figure 3 nanomaterials-12-04488-f003:**
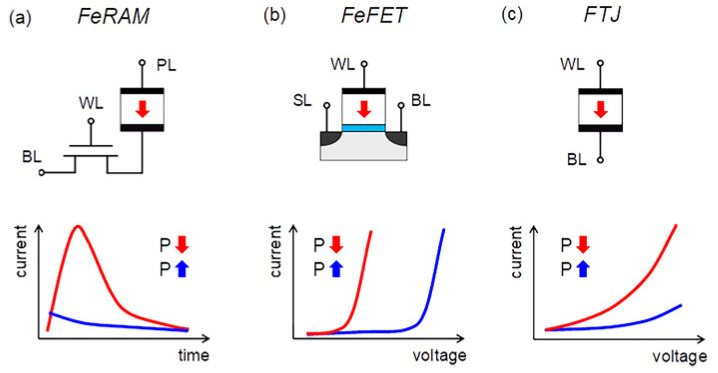
(**a**) FeRAM, (**b**) FeFET, and (**c**) FTJ [34].

**Figure 4 nanomaterials-12-04488-f004:**
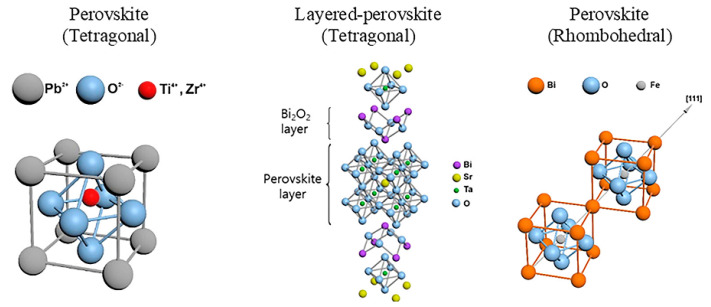
Classic ferroelectric materials: Pb(Zr,Ti)O_3_, SrBi_2_Ta_2_O_9_, and BiFeO_3_ [34].

**Figure 5 nanomaterials-12-04488-f005:**
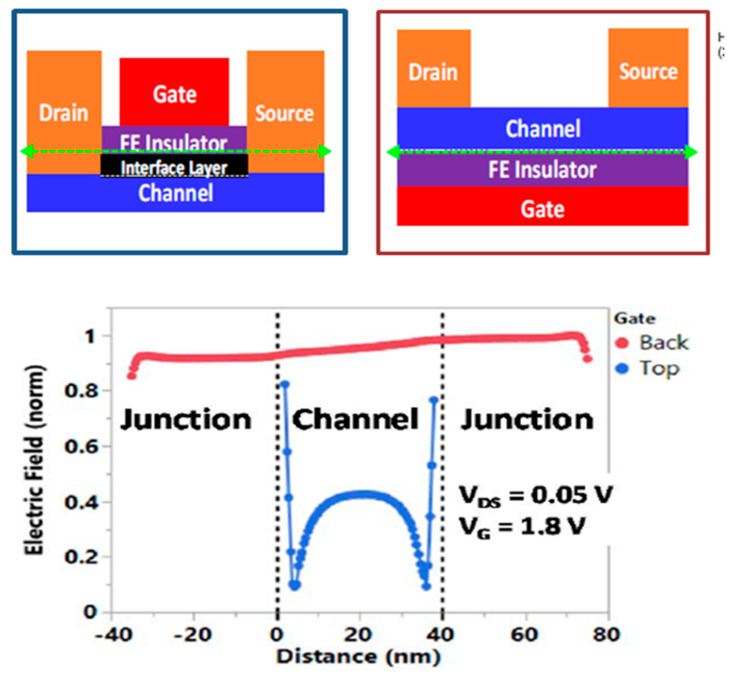
FeFET design change to improve field uniformity in ferroelectric along the channel. Copyright © 2020, IEEE [99].

**Figure 6 nanomaterials-12-04488-f006:**
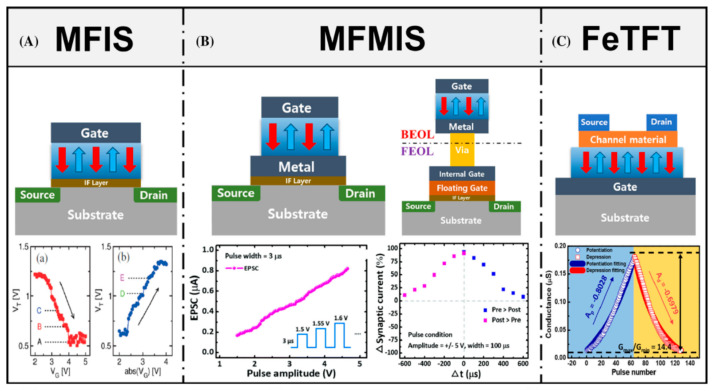
Various ferroelectric devices with a three-terminal structure (**upper** row) and corresponding characteristics of synaptic plasticity (**lower** row). (**A**) MFIS, (**B**) (**left**) MFMIS and (**right**) MFMIS with the MFM capacitor integrated into the BEOL and the underlying MOSFET integrated into the FEOL, and (**C**) FeTFT. Schematic of FeFETs with different channel geometries and synaptic potentiation (**a**) or depression (**b**) [73].

**Figure 8 nanomaterials-12-04488-f008:**
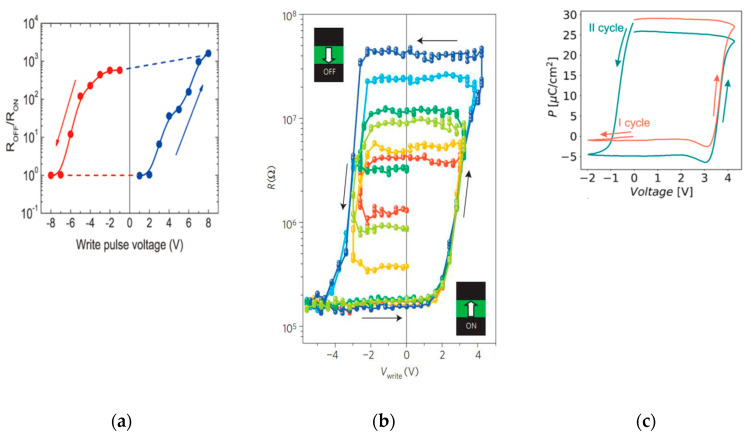
(**a**–**c**) An example of logarithmic R_OFF_/R_ON_ and polarization, respectively, as a function of switching voltage Copyright © 2012, 2019 American Chemical Society [122,123,124].

**Figure 9 nanomaterials-12-04488-f009:**
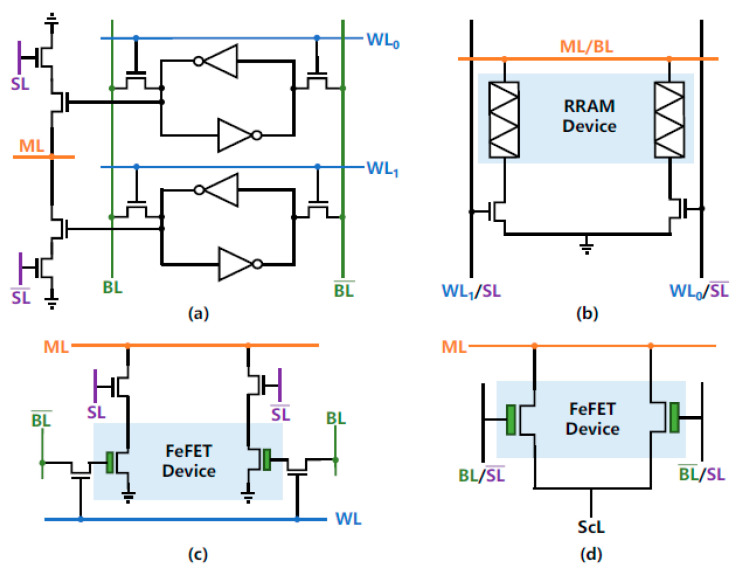
TCAM cell designs based on: (**a**) CMOS static random access memory required 16 transistors, BL, bit line, ML, pre-charged matchline, SL, search line, WL, write line, (**b**) based on resistive storage elements, (**c**) based on four CMOS FET and two FeFETs, (**d**) based on two FeFETs [https://arxiv.org/abs/2101.06375 Access Date: 10 December 2022].

**Table 1 nanomaterials-12-04488-t001:** Comparison of ferroelectric materials: PZT, SBT, BFO, doped HfO_2_, Al_x_Sc_1−x_N [34].

Ferroelectrics	Pb(Zr,Ti)O_3_	SrBi_2_Ta_2_O_9_	BiFeO_3_	Doped HfO_2_ Hf_x_Zr_1−x_O_2_	Al_x_Sc_1−x_N_8_
Pr (μC/cm^2^)	10–40	5–10	90–95 (along [111])	10–40	80–110
Ec (kV/cm)	50–70	30–50	100–1500	800–2000	2000–5000
*ε* _0_	∼400	∼200	∼50	∼30	∼25
Endurance (cycles)	>1 × 10^15^ on oxide electrode	Good on Pt electrode	Good on oxide electrode	>1 × 10^11^ on TiN	>1 × 10^5 a^
Min. physical thickness (nm)	50			<5	<50 ^a^
Crystallization temperature (°C)	Low	High	Low	400–800	300–400
Curie temperature (°C)	∼400	∼400	∼700	0–500	>600

^a^ Early results, improvements expected.

**Table 2 nanomaterials-12-04488-t002:** Key characteristics according to representative polymorphs of fluorite-structured HfO_2_ and ZrO_2_ [73].

	Monoclinic	Orthorhombic	Tetragonal
Crystal structure 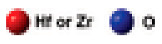	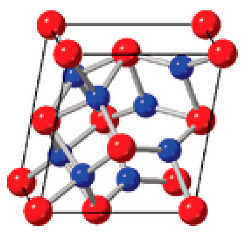	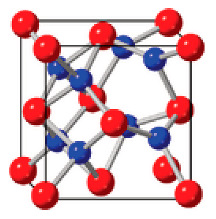	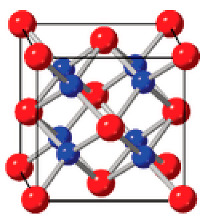
Property	Dielectric	Ferroelectric	Dielectric
Polarization	0	*P*_r_ ~ 50–60 μC cm^−2^	0
Dielectric constant	19–25	24–29	24–57
Space group	*P*2_1_/c	*P*ca2_1_	*P*4_2_/nmc

**Table 3 nanomaterials-12-04488-t003:** TCAM cell parameters [26].

	16T CMOS	2T-2R	2FeFET
Cell area (μm^2^)	1.12	0.28	0.15
Write energy (fJ per bit)	4.8	720	1.4

## Data Availability

Not applicable.

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
