# Peer review of "Ferroelectric Devices for Content-Addressable Memory"

_nanomaterials, 2022, doi:10.3390/nano12244488_

Round 1

Reviewer 1 Report

The paper is good for the technology review, however, it is a bit short on the CAM review and analysis.  For example, actual area comparison values for the various CAM cells in figure 9. would be good.  

Also, Table 1 needs some better description in the text - particularly the row labeled @0.

Author Response

A comparison of various CAM cells is presented in Table 3.

Table 1 has been edited.

Reviewer 2 Report

Nice review. I have some suggestions as listed following.

S1: Most contents discussed the ferroelectric devices, and only a small part of the manuscript discussed the CAM. So I suggest changing the title.

S2: I recommend adding some recent work.

S3: The “STP” on page 6 is a typo, please check the full manuscript carefully.

Author Response

S1: The title of the article has been changed to “Ferroelectric devices for content-addressable memory”.

S2: We think that the new results are well reflected in the article.

S3: Typos on page 6 corrected.
